# Effect of Mo Content on Hydrogen Evolution Reaction of 1400 MPa-Grade High-Strength Bolt Steel

**DOI:** 10.3390/ma16031020

**Published:** 2023-01-22

**Authors:** Xilin Xiong, Keke Song, Jinxu Li, Yanjing Su

**Affiliations:** 1Beijing Advanced Innovation Center for Materials Genome Engineering, Beijing 100083, China; 2Corrosion and Protection Center, University of Science and Technology Beijing, Beijing 100083, China; 3Department of Physics, University of Science and Technology Beijing, Beijing 100083, China

**Keywords:** hydrogen evolution reaction, Mo content, first-principle calculation, hydrogen permeation

## Abstract

The effect of Mo content of 1400 MPa-grade high-strength bolt steel on hydrogen diffusion behavior and the hydrogen evolution reaction were studied using a hydrogen permeation experiment, potentiodynamic polarization tests, thermal desorption spectroscopy, and the first-principle calculation. Two 1400 MPa-grade high-strength bolt steels with different Mo content were used. Based on the potentiodynamic polarization tests, both steels’ electrochemical behavior was similar in the test range. The hydrogen permeation experiment showed that the process of hydrogen adsorption and absorption was significantly promoted, and hydrogen desorption and recombination were slightly promoted, with the Mo content increasing from 0.70 to 1.09 wt%. The thermal desorption spectroscopy showed the overall reaction of hydrogen permeation and evolution. The increasing Mo content facilitated hydrogen entry behavior and increased the hydrogen content. According to the first-principle calculation and the density functional theory, this phenomenon is induced by the stronger bonding ability of Mo-H than Fe-H. This work could guide the design of 1400 MPa-grade high-strength bolt steel.

## 1. Introduction

According to the national standard for bolt steel GB/T 3098.1-2010, steel with a strength stronger than 1200 MPa-grade has not been considered [1]. Due to the practical needs of lightweight vehicles, it is exceptionally urgent to develop 1400 MPa-grade high-strength bolt steel. However, metallic materials’ hydrogen embrittlement (HE) is modest in modern industry, especially in high-strength steel [2,3]. An efficient approach to increasing steel’s resistance to HE is by adding alloying elements [4,5,6,7]. The alloy element’s content influences hydrogen entry behavior, which is the first step of HE and will eventually affect the resistance of HE [8,9]. Therefore, investigating the relationship between the entrance of hydrogen and alloy elements’ content is of great significance.

Since Mo can form carbide precipitation as a strong hydrogen trap in steels, it is regarded as an alloying element, which can improve the resistance of HE [10]. Mo carbide precipitation enhances HE resistance and boosts strength through precipitation hardening [11]. According to modeling performed by Yamasaki et al., the major features that increase Mo_2_C’s resistance to HE are the carbide size and number density [12]. Lee et al. examined the function of Mo/V carbides in the tempered-martensitic-steel HE resistance. The Mo steel had strong HE resistance owing to the Mo carbides [13]. However, Wei et al. showed that the various carbides’ capacity to trap hydrogen evolved in increasing order, from Mo_2_C to VC to TiC to NbC, showing that Mo_2_C was the least potential carbide to strengthen resistance to HE [14]. Few studies have focused on the influence of Mo on the entrance of hydrogen, even though it is the first step of HE.

With the development of hydrogen permeation models and experiments, the kinetic parameters for the hydrogen evolution reaction (HER) can be determined. In acid solution, the HER and hydrogen diffusion are shown as below [15]: (1)$$H^{+}+e^{-}+M\overset{k_{1}}{\to }M-H_{ads}$$
where $M-H_{ads}$ denotes a hydrogen atom that has been adsorbed on a metal surface, and $k_{1}$ is the rate constant of the discharge step of the HER. Through the Tafel chemical recombination mechanism, a hydrogen molecule is created when some of the adsorbed hydrogen atoms unite again,
(2)$$M-H_{ads}+M-H_{ads}\overset{k_{2}}{\to }2M+H_{2}(g)$$
where $k_{2}$ is the rate constant of the recombination step of the HER. Some hydrogen atoms that have been adsorbed are then absorbed. The following is how hydrogen atoms spread into the area beneath the metal surface:(3)$$M-H_{ads}\overset{}{↔}H_{abs}+M$$
where $H_{abs}$ represents the absorbed hydrogen atom within the metal lattice.

Based on the Devanathan–Stachurski double-cell hydrogen permeation experiment [16], in order to assess the rate constants of the HER and surface coverage quantitatively, Iyer et al. presented the Iyer–Pickering–Zawenzaden (IPZ) model [15,17,18]. The effect of Mo on the hydrogen entrance behavior can be characterized based on the IPZ model, shown in Equations (4)–(8).
(4)$$i_{c}\exp\left(\frac{F}{RT}\alpha \eta \right)=i_{0}-\frac{i_{0}}{Fk}i_{\infty }$$
(5)$$i_{\infty }=k\sqrt{\frac{F}{k_{2}}}\sqrt{i_{r}}$$
(6)$$i_{0}=Fk_{1}C_{H^{+}}$$
(7)$$i_{\infty }=Fk\theta$$
(8)$$\frac{1}{k}=\frac{1}{k_{abs}}+\frac{k_{des}}{k_{abs}}\frac{L}{D}$$
where $i_{r}$ and $i_{\infty }$ are the adsorbed-hydrogen-atom-recombination current density and the hydrogen permeation steady-state current density; $i_{c}$ represents the charging-current density, $i_{c}=i_{\infty }+i_{r}$; α is the transfer coefficient; *T* is temperature; *F* is Faraday’s constant; *R* is the gas constant; $C_{H^{+}}$ represents the H^+^ concentration; η is the overpotential (η
*= E_applied_* − *E_corr_; E_applied_* represents the applied cathodic potential; *E_corr_* represents the corrosion potential); $i_{0}$ represents the exchange current density; θ is the hydrogen surface coverage; and $k_{abs}$ and $k_{des}$ are the forward and backward rate constants of Reaction (3). *D* is the hydrogen diffusion coefficient in the steel membrane; the sample thickness of hydrogen permeation is *L*. The coupled discharge–recombination mechanism is followed by the HER when an appropriate *E_applied_* for hydrogen charging is selected [19]. Thus, the linear relationship between $i_{\infty }$ and $\sqrt{i_{r}}$, $i_{c}\exp\left(\frac{F}{RT}\alpha \eta \right)$ and $i_{\infty }$ can be obtained. The rate constants of the HER and $\theta_{H}$ can be calculated based on Equations (4)–(8).

Based on the examination of the double cell electrochemical hydrogen permeation and IPZ model, this paper examines the impact of Mo content on the HER and hydrogen diffusion of 1400 MPa-grade high-strength bolt steel. The results were validated using density functional theory (DFT) simulation and thermal desorption spectroscopic characteristics.

## 2. Experimental Methods

### 2.1. Tested Materials

1400 MPa-grade high-strength bolt steels with different Mo contents were used, and its composition is shown in Table 1. In the hydrogen permeation tests, the test samples were round, with a 12 mm diameter. The thickness of the three samples for 1# steel are 0.37 mm, 0.61 mm, and 0.89 mm, respectively. The thickness of the three samples for 2# steel are 0.36 mm, 0.61 mm, and 0.89 mm, respectively. Each sample’s surface was polished using automatic polishing equipment with sandpaper with a maximum grit of 1500 and a polishing cloth. Sputter deposition was used to deposit 100 nm of nickel on one side of the samples while they were under an extremely high vacuum. A thin-film sputtering system was used to carry out the sputter deposition (LAD18, KJLC). In the potentiodynamic polarization tests, the sample size was 10 × 10 × 5 mm, and the surface exposure to the solution was 10 × 10 mm. This surface was also polished with 1500-grit sandpaper and a polishing cloth. The other areas of the samples were coated with epoxy. 

### 2.2. Electrochemical Investigation Methods

The potentiodynamic polarization tests were performed in 0.05 mol/L H_2_SO_4_ at 25 °C by Gamry interface 1000 after the open circuit potential (OCP) reached the steady state. A three-electrode system was used in the electrochemical cell. As the counter and reference electrodes, respectively, a platinum plate and saturated calomel electrode (SCE) were chosen. The samples’ potential was changed from an initial potential of −300 mV vs. *E*_OCP_ to 300 mV vs. *E*_OCP_ at a rate of 0.33 mV/s.

The Devanathan–Stachurski double cells were used to characterize the hydrogen permeation behavior and HER of two kinds of steels [16]. The three-electrode system was used in each cell. The hydrogen exit side of the specimens covered with nickel film was immersed in 0.2 mol/L NaOH solution. *η* applied on this side was +250 mV. The hydrogen entry side was immersed in 0.05 mol/L H_2_SO_4_ solution. Constant *E_applied_* was applied to the samples at the hydrogen entry side. *E_applied_ = η + E_corr_*; *η* ranged from −700 mV to −620 mV in 20 mV increments. Once $i_{\infty }$ reached a steady state, *E_applied_* changed to another value. Thus, $i_{c}$, $i_{\infty }$ and $i_{r}$ under each *η* could be obtained.

### 2.3. Thermal Desorption Spectroscopy (TDS)

The hydrogen content in the specimens was calculated by the results of TDS. A solution of 3% NaCl + 0.3% NH_4_SCN was used during the 72 h hydrogen-charging process, and the hydrogen-charging-current densities were 0.1 and 1 mA/cm^2^. Before the specimens were placed in the vacuum tube, they were cleaned and dried with deionized water and ethanol (within 5 min). The sample was cylinder-shaped with a size of Φ 5 × 25 mm. The test was started when the vacuum level was sufficiently low. The heating rate and final temperature were 100 K h^−1^ and 800 k, respectively.

### 2.4. Computational Details and Structural Models

The Vienna ab initio simulation package (VASP) was used for our calculations [20,21] using the Perdew–Burke–Ernzerhof generalized gradient approximation exchange-correlation density functional [22]. We used a (4 × 4 × 1) k-mesh within the Monkhorst–Pack scheme and a plane-wave energy cutoff of 450 eV for all calculated systems. In this paper, a 3 × 3 Fe (100) surface with a thickness of 9 atomic layers was selected to simulate the adsorption of H on the surface, as shown in Figure 1. Because the Fe (100) surface was more open compared to the Fe (110) surface. The bottom three-layer atoms of the surface model were fixed in their perfect bulk lattice sites, while other Fe atoms, doped-alloy atoms (Mo) and H atoms were free to relax. The adsorption energy of the H atom on the study surface can be expressed as:(9)$$∆E=E\left({Fe}_{x}A_{y}H\right)-E\left({Fe}_{x}A_{y}\right)-\frac{1}{2}E\left[H_{2(g)}\right]$$
where $E\left({Fe}_{x}A_{y}H\right)$ represents the system’s total energy. The system contains an *H* atom, *x Fe* atoms, and *y Mo* atoms. $E\left({Fe}_{x}A_{y}\right)$ is the system’s total energy of *x Fe* atoms and *y Mo* atoms without *H* atoms. $E\left[H_{2(g)}\right]$ represents one-half of the total energy of gaseous H_2_ molecules. According to the first-principles calculations, the total energy of the H_2_ molecule was −6.76 eV, and the bond length was 0.751 Å. These values were well-concordant with the earlier published results [8,23,24].

## 3. Results 

### 3.1. The Characteristic of Hydrogen Diffusion and the HER

The Tafel curves were measured on the surface of two kinds of steels, as shown in Figure 2. The corrosion potential *E_corr_* is −0.33 *V_RHE_* for both of 1# and 2# steels. In the Tafel linear area, overpotential *η* and the cathodic polarization current *i* show the relationship as Equation (10) [25]:(10)$$\eta =-\frac{2.3RT}{\alpha F}logi_{0}+\frac{2.3RT}{\alpha F}logi$$

The transfer coefficient α can be calculated based on Equation (10) and Figure 2. α is 0.43 and 0.42 for 1# and 2# steel, respectively.

During the hydrogen permeation tests, $E_{applied}=-700mV+E_{corr}$ was applied on the hydrogen entry side. Once the permeate flux reached a steady state, $E_{applied}$ increased +20 mV. The results of hydrogen permeation of 1# and 2# steel with different thicknesses are shown in Figure 3. One can obtain $i_{c}$, $i_{\infty }$, $i_{r}$ and the diffusion coefficient D, which are shown in Table 2. D was calculated using the time-lag method, as shown in Equation (11) [16].
(11)$$D=\frac{L^{2}}{15.3t_{b}}$$
where *L* represents the specimens’ thickness; $t_{b}$ is the breakthrough time of the hydrogen permeation test. It can be observed that D increased with the increase in the thickness of the specimens. This result corresponds with previous works [26,27]. According to Zhang et al., the thickness effect is produced by the absorption and desorption processes, in which hydrogen atoms must overcome energetic barriers in order to enter or exit the sample [26]. Additionally, Figure 3d reflects the thickness effect. The value of $i_{\infty }$ increased with the decrease in the sample’s thickness. However, $i_{c}$ was almost independent of the sample’s thickness based on Figure 3c because $i_{c}=i_{\infty }+i_{r}$, and $i_{\infty }$ is several magnitudes smaller than $i_{r}$, while $i_{r}$ only matters with the surface of the hydrogen entry side. It can be found that with the increasing Mo content, both $i_{c}$ and $i_{\infty }$ increased.

Figure 4a shows the relation between $i_{c}\exp\left(\frac{F}{RT}\alpha \eta \right)$ and $i_{\infty }$. Similarly, Figure 4b shows the relation between $i_{\infty }$ and $i_{r}^{1/2}$. Within these figures, data points under a single experimental condition keep a linear relationship. This phenomenon shows that the mechanism of the HER is the coupled Volmer discharge–Tafel recombination. *k* can be estimated from the slope of Equation (4) and the data in Figure 4a. *k*_1_ can be calculated by Equation (6) based on the intercept obtained in Figure 4a. *k*_2_ can be obtained by Equation (5) and the data in Figure 4b. The forward and backward rate constants of Reaction (3), $k_{abs}$ and $k_{des}$, can be obtained by fitting the data in Figure 4c with Equation (8). These results are shown in Table 2. The hydrogen surface coverage θ can be obtained according to Equation (7). As we can see from Figure 4d, θ was independent from the samples’ thickness. The increase in Mo content increased θ. 

### 3.2. The Characteristic of Hydrogen Desorption Behavior

Figure 5 shows the hydrogen desorption rate curves of original specimens and specimens after hydrogen charging in 3% NaCl + 0.3% NH_4_SCN for 72 h. The current densities of 0.1 and 1 mA/cm^2^ were chosen in the hydrogen-charging experiment to verify the influence of Mo content on hydrogen entry behavior under different conditions. As for the hydrogen-charged 1# steel, one significant low-temperature peak showed at 164 °C for the 0.1 mA cm^−2^ charged specimen and 173 °C for the 1 mA cm^−2^ charged specimen. As for the hydrogen-charged 2# steel, one peak showed at 188 °C for 0.1, and the 1 mA cm^−2^ charged specimens. Additionally, a shoulder peak appeared at ~150 °C. The total hydrogen concentration can be calculated by the integral area under the peaks. Under each hydrogen-charging-current density, it is seen that the hydrogen content marginally increased with the increase in Mo content. 

### 3.3. The First-Principal Calculation of Hydrogen Adsorption

In this work, the adsorption behavior of H on a pure Fe (100) surface was tested, and the hydrogen adsorption energies on the on-top site, twofold bridge site, and fourfold hollow site on the Fe (100) surface were calculated. The adsorption energy of the fourfold hollow site was the lowest at −0.386 eV, and the distance between the adsorbed H atoms and the surface was the smallest (0.385 Å). These findings corroborate earlier experimental and computational results [23,28,29,30]. However, the adsorption energy of H on the twofold bridge sites on the Fe (100) surface was −0.380 eV, which was almost equal to that of the fourfold hollow sites. Then, the effect of surface doping with Mo on the H adsorption behavior is calculated in this paper. A Mo replaces a Fe atom in the first layer of the surface. In order to highlight the effect of Mo atoms on H adsorption, the adsorption energies of H on different adsorption sites within a specific range around the doped alloy atoms were calculated, and the most stable chemisorption configuration on the doped surface was selected from them. The computational results show that H was most stable near Mo doping at the twofold bridge position. The adsorption energy was −0.498 eV. To investigate the effect of Mo concentration on H on the doped surface, we calculated the effect of two Mo atoms doping on the H adsorption energy. Since the doping of two Mo atoms did not change the H adsorption site, to study the impact of two Mo atoms on the H adsorption, the Mo doping site is shown in Figure 1; H was adsorbed on the twofold bridge composed of two Mo atoms, and the adsorption energy of H at this position was −0.778 eV. These calculation results show that H atoms adsorbed on the surface more stably with Mo doping than on the pure Fe (100) surface. The increased Mo doping concentration made the H atoms’ trapping ability stronger. 

To deeply explore the mechanism of H adsorption by Mo atoms, we calculated the charge density maps of the twofold bridge adsorption sites of H on Fe- and Mo-doped (100) surfaces with different concentrations, respectively, as shown in Figure 6. Comparing the electronic structures of (a) and (b) in Figure 6, it can be found that the doping of Mo increased the charge density on the surface. As can be observed, Mo-H had a stronger connection than Fe-H because the charge density between the two atoms was higher. It is obvious from Figure 6b that the doping of Mo also reduced the adsorption of Fe to H. With the increase in Mo doping concentration, it can be seen from Figure 6c that the charge density between Mo-H after two Mo atoms doping was slightly smaller than that of one Mo atom doping, stronger than the pure Fe system. In particular, Mo-H-Mo had a stronger charge density than pure Fe and a Mo-doped system. In addition, it can be seen that the charge density between Mo-Fe was stronger than Fe-Fe and that Mo doping increased the surface and subsurface charge density. The mechanism wherein Mo has a solid ability to adsorb H from the electronic configuration, and the doping of Mo can enhance the bonding ability near the surface, was revealed.

## 4. Discussion

In this work, two types of steels with different Mo content were used to study the influence of Mo content on the HER and hydrogen permeation behavior. In the hydrogen permeation tests, for each steel, samples with three kinds of thicknesses were used. Thus, the value of k and D under different sample thicknesses can be obtained. Based on Equation (8), the value of $k_{abs}$ and $k_{des}$ can finally be calculated. 

Table 2 shows that all the rate constants were independent of the specimens’ thickness. It means that the rate constants are the intrinsic properties of materials. It can be observed that the value of $k_{1}$, $k_{2}$, $k_{abs}$ and $k_{des}$ increased by varying degrees because of Mo addition; the increase rates are shown in Figure 7. 

According to the first-principle calculation results, the adsorption energy decreased from −0.386 eV to −0.778 eV with the increased Mo surface doping due to the stronger bonding ability of Mo-H than Fe-H. Thus, the forward reaction of the hydrogen adsorption reaction, Reaction (1), was promoted and θ increased. 

Additionally, it can be found that the forward and backward rate constants of Reaction (3), $k_{abs}$ and $k_{des}$, increased with Mo content. The increase in $k_{abs}$ and $k_{des}$ means that the movement of hydrogen atoms between the surface and subsurface was intensified. The value of $k_{abs}$ increased by 170% and $k_{des}$ increased by 70%, which means increasing Mo content promoted the forward reaction of Reaction (3) and facilitated the hydrogen entry behavior on the whole. However, due to the increase in $k_{des}$ and θ, more hydrogen atoms adsorbed on the surface of specimens. Thus, $k_{2}$ increased by 9%, and Reaction (2) was slightly promoted.

Since Reaction (1) and (3) were significantly enhanced, the entry behavior of hydrogen atoms was facilitated with the increase in Mo content. It can be verified by the results of TDS, as shown in Figure 5. Under the same hydrogen-charging condition, more hydrogen atoms permeated into the samples.

Thus, combining hydrogen permeation tests and the potentiodynamic polarization with the IPZ model, the value of HER kinetic parameters, $k_{1}$, $k_{2}$, $k_{abs}$ and $k_{des}$, for 1# and 2# steels were determined. Using the first-principle calculation and TDS, the accuracy of these kinetic parameters was verified. In conclusion, increasing Mo content facilitated hydrogen entry behavior and increased the hydrogen content in 1400 MPa-grade high-strength bolt steel. 

## 5. Conclusions

With the Mo content of 1400 MPa-grade high-strength bolt steel increasing from 0.70 to 1.09 wt%, the effect of Mo content on hydrogen diffusion and the HER was systematically studied. 

(1)With increased Mo content, the hydrogen-charging-current density and steady-state current density increased, while the hydrogen diffusion coefficient decreased.(2)Due to the stronger bonding ability of Mo-H than Fe-H, the adsorption energy decreased with the increase in Mo content. It induced the promotion of the hydrogen adsorption reaction and increased hydrogen surface coverage.(3)The increase in Mo content slightly promoted hydrogen recombination and significantly advanced the forward reaction of hydrogen absorption and desorption.(4)On the whole, increasing the Mo content facilitated hydrogen entry behavior and increased the hydrogen content in 1400 MPa-grade high-strength bolt steel.

## Figures and Tables

**Figure 1 materials-16-01020-f001:**
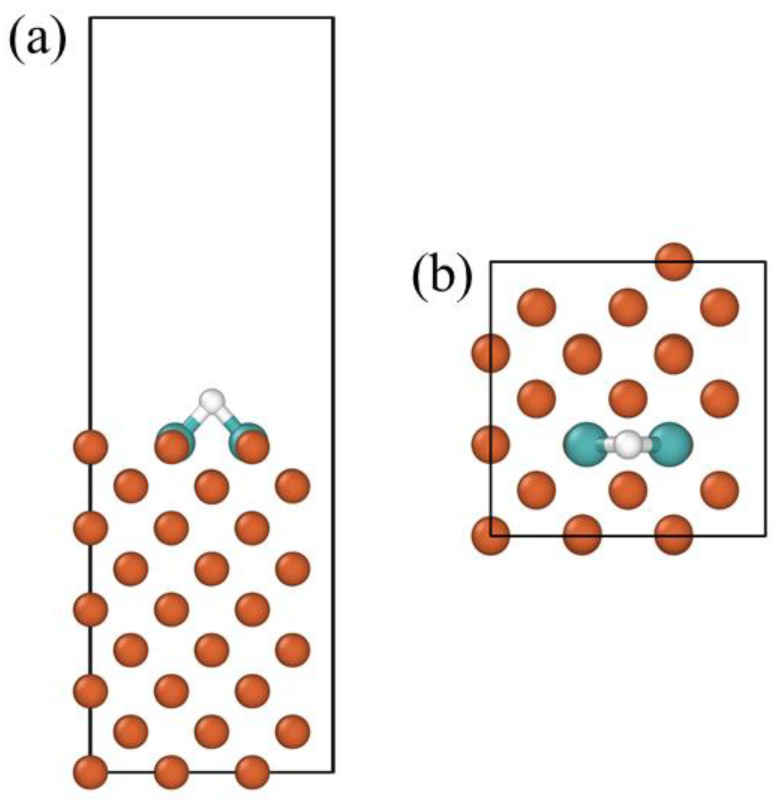
H adsorption sites on Mo-doped Fe (100) planes. (**a**) The front view, (**b**) The plan view. The green one is the Mo atom, the white one is the hydrogen atom, and the others are the Fe atom.

**Figure 2 materials-16-01020-f002:**
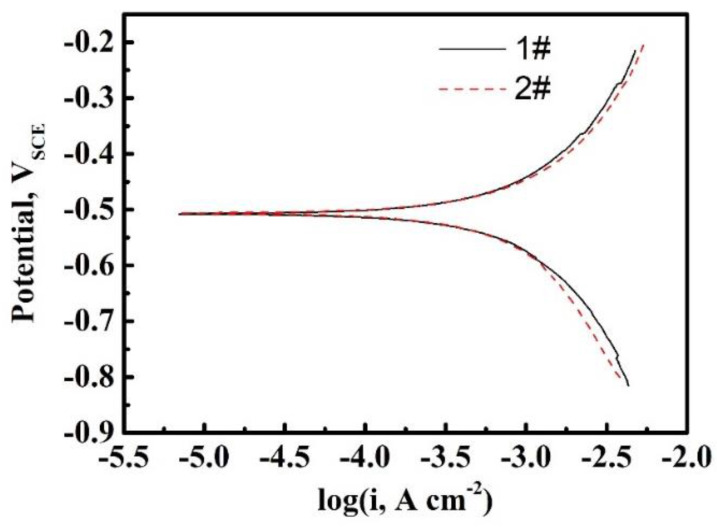
The potentiodynamic polarization of 1# and 2# steel in 0.05 mol/L H_2_SO_4_ at 25 °C.

**Figure 3 materials-16-01020-f003:**
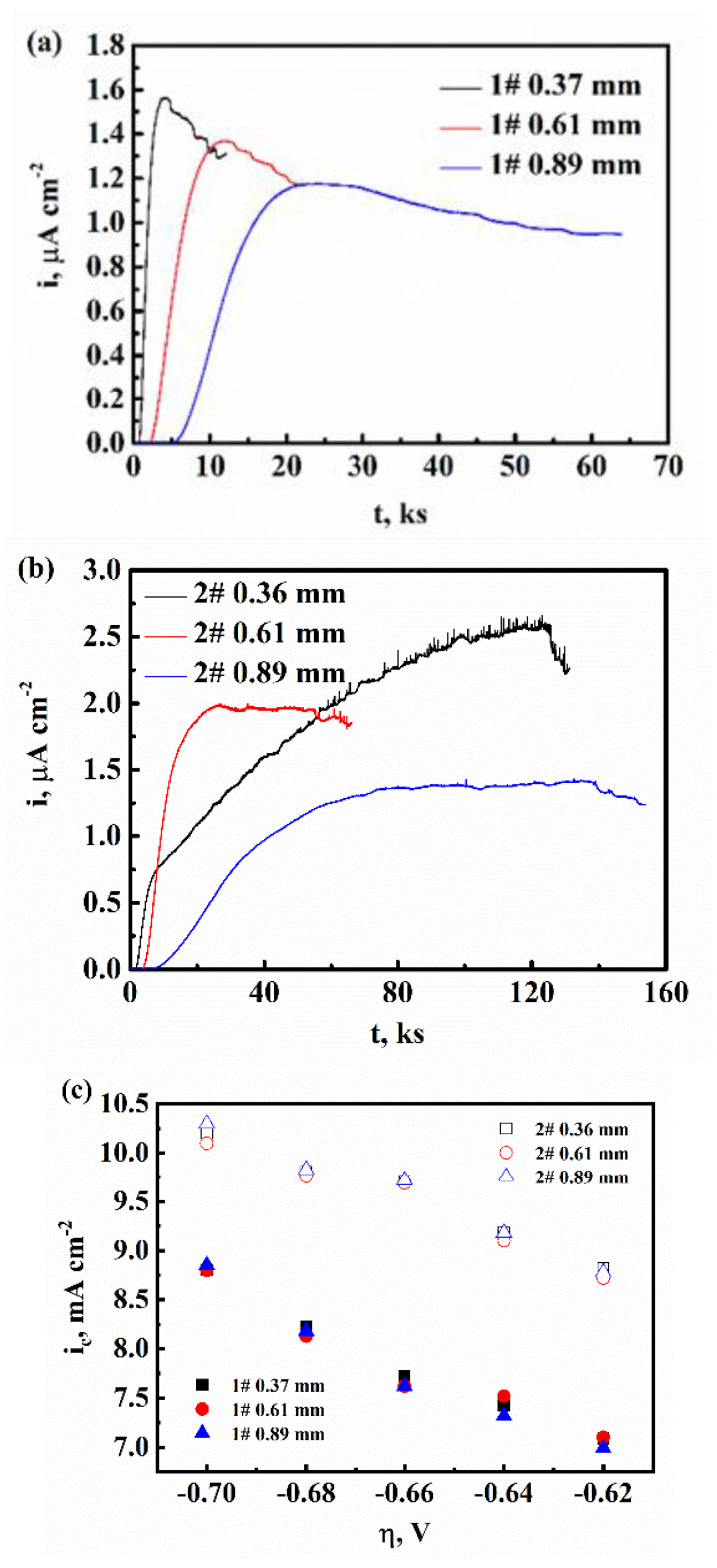
The hydrogen permeation tests of 1# and 2# steel with different thicknesses obtained in 0.05 mol/L H_2_SO_4_ at 25 °C: (**a**) The hydrogen permeation curve of 1# steel with different specimen’s thickness; (**b**) The hydrogen permeation curve of 2# steel with different specimen’s thickness; (**c**) The relation between *i_c_* and *η*; (**d**) The relation between *i_∞_* and *η*.

**Figure 4 materials-16-01020-f004:**
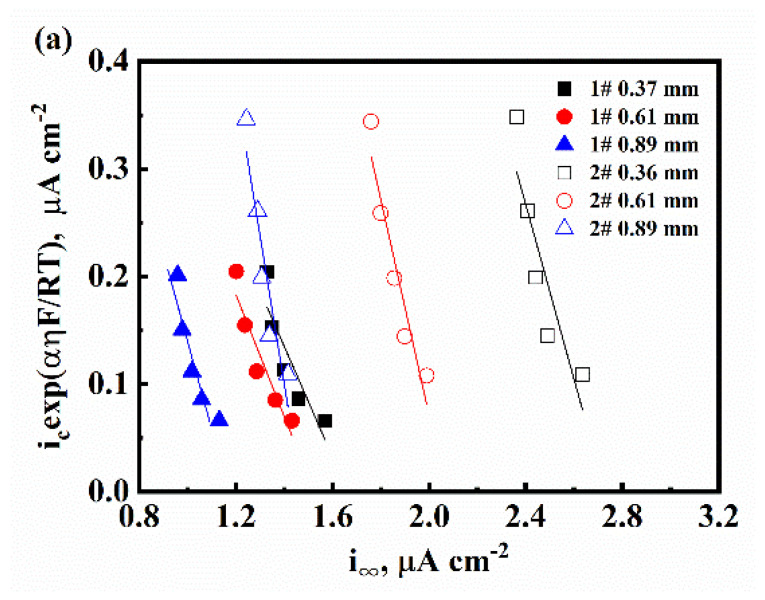
The characteristic of HER rate constants based on the IPZ model: (**a**) The relation between $i_{c}\exp\left(\frac{F}{RT}\alpha \eta \right)$ and $i_{\infty }$, the fitting results using Equation (4) are represented by the solid lines; (**b**) The relation between $i_{\infty }$ and $i_{r}^{1/2}$, the fitting results using Equation (5) are represented by the solid lines; (**c**) The relation between 1/*k* and the specimens’ thickness *L*; the solid lines are the fitting results based on Equation (8); (**d**) The relation between the hydrogen surface coverage θ and the cathodic overpotential *η*.

**Figure 5 materials-16-01020-f005:**
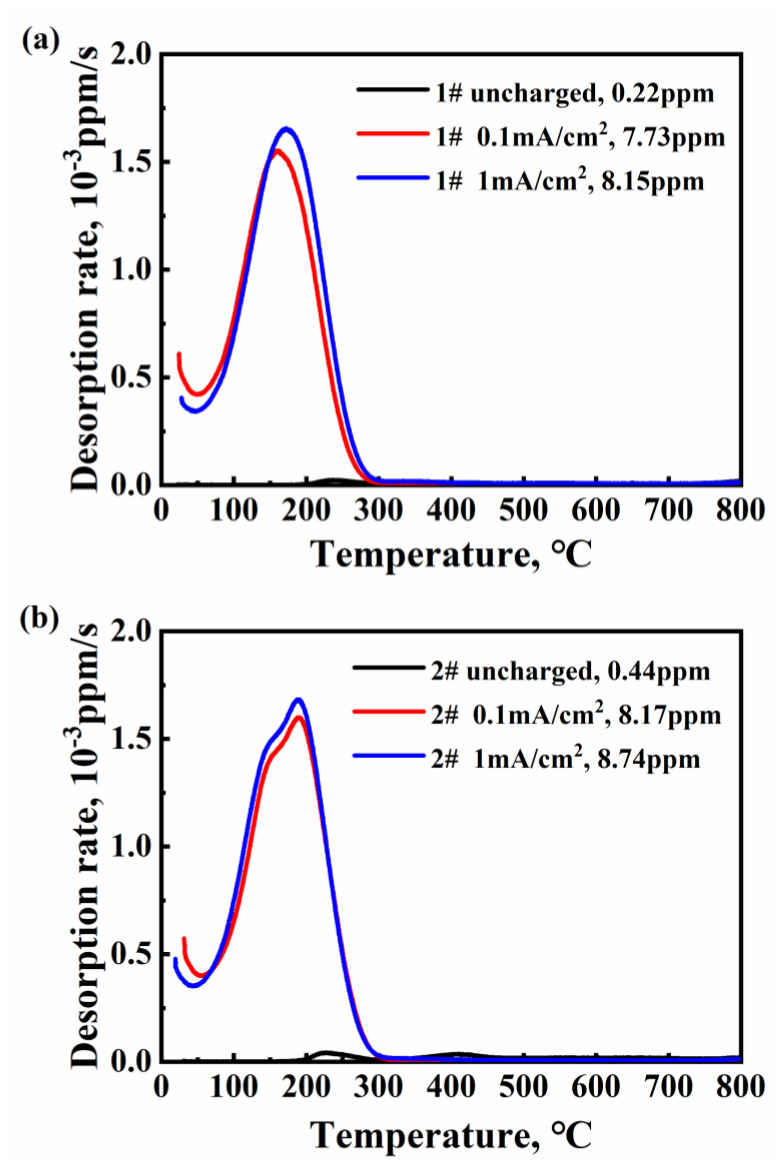
The effect of Mo content on the TDS spectrum of 1400 MPa-grade high-strength bolt steel. The heat rate was 100 K h^−1^. (**a**) TDS spectra for 1# steel with different hydrogen-charging-current densities, (**b**) TDS spectra for 2# steel with different hydrogen-charging-current densities.

**Figure 6 materials-16-01020-f006:**
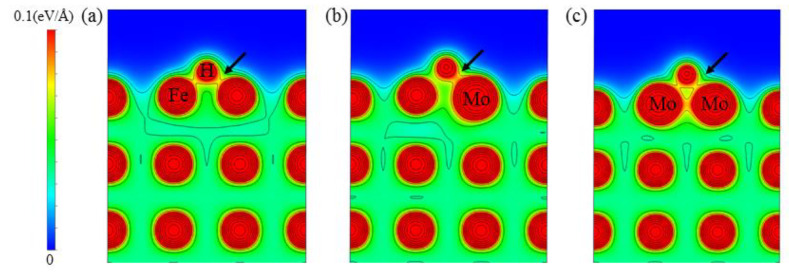
H adsorption charge density map on the (100) surface. (**a**) H adsorption on a pure Fe (100) surface; (**b**) H adsorption on a Mo-doped surface; (**c**) H adsorption on surface doped by two Mo atoms.

**Figure 7 materials-16-01020-f007:**
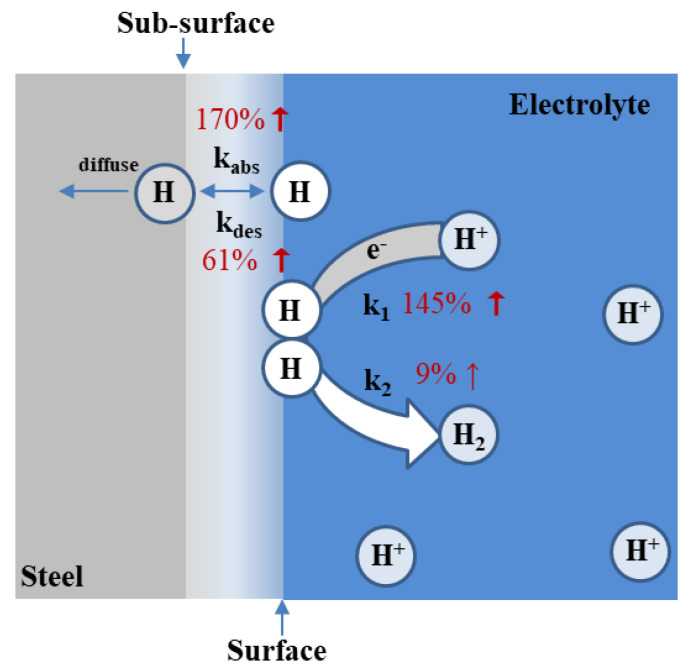
The schematic of Mo addition effect on HER.

**Table 1 materials-16-01020-t001:** The composition of 1400 MPa-grade high-strength bolt steel (mass, wt%).

	C	Si	Mn	Cr	Mo	V
1#	0.41	0.22	0.31	1.00	0.70	0.34
2#	0.42	0.19	0.31	1.00	1.09	0.35

**Table 2 materials-16-01020-t002:** Parameters of the hydrogen permeation experiments and the IPZ model.

Sample	*L*, mm	*D*, ×10^7^ cm^2^/s	*k*, ×10^−11^ mol cm^−2^ s^−1^	*k_1_*, ×10^−1^ cm s^−1^	*k_2_*, ×10^−7^ mol cm^−2^ s^−1^	*k_abs_*, ×10^−11^ mol cm^−2^ s^−1^	*k_des_*, ×10^−6^ cm s^−1^
1#	0.37	1.41	1.72	0.89	1.11	2.47	1.59
	0.61	1.50	1.58				
	0.89	1.51	1.24				
2#	0.36	0.59	2.83	2.18	1.21	6.67	2.56
	0.61	0.68	2.15				
	0.89	0.78	1.53				

## Data Availability

The raw/processed data required to reproduce these findings cannot be shared at this time as the data also form part of an ongoing study.

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
