# Peer review of "Effect of Mo Content on Hydrogen Evolution Reaction of 1400 MPa-Grade High-Strength Bolt Steel"

_materials, 2023, doi:10.3390/ma16031020_

Round 1
Reviewer 1 Report
Notes:
1- It is not preferable to use abbreviated terms in the abstract, for example (HER) without a full definition.
2- It is assumed in the abstract to indicate the most important result that was at any condition.
3- No reference was made in the abstract to the adoption of DFT as a means of validation of results.
4- What is the difference between #1 & #2 samples, in the absence of Table No. (1) that is supposed to be mentioned in the experimental part .
5- As indicated in Paragraph 2.1, samples 1# and 2# steel are approximately the same thickness.
6- It was mentioned in Paragraph 2.1 (Steel with different Mo contents were used, and its composition is shown in Table 1).There is no table number 1 in the paper that should be added for the purpose of comparison and discussion.
7- The values of the TDS variables and the rest of the tests have not been clarified on what basis they were determined (selected).
8- The author (s) used the term (We) a lot, and this is not preferred in writing papers.
9- From Figure (3) the obvious difference in hydrogen permeation between #1 & #2 # steel samples needs further discussion.
10- The discussion in general needs to reinforce and explain the behavior and explain why two types of samples #1 and #2 were chosen, with no clear difference between the two groups.
11-The conclusions are good.
12- The references are well related to the subject of the paper.
Reviewer 2 Report
1. Define HER in the abstract
2. Symbols used like that 1#, what the author conveyed by this symbols
3.Paper seems good, need to give proper definition of the above points
Reviewer 3 Report
The authors reported the effect of Mo content on the hydrogen evolution reaction of 1400 2MPa-grade high-strength bolt steel. The manuscript is not well written and misses the tables, moreover, the provided results and discussion are not enough; the following are some of my comments:-
- The Abstract is too short and has not provided enough information on how the authors claimed their findings. Why the corrosion measurements are not mentioned?
- What is the meaning of HER in the Abstract? Authors should write the full name before putting its abbreviation when it is mentioned for the first time. Not everyone knows that HER means hydrogen evolution reaction!!!!
- Experimental methods, page 3, line 82 and line 83, “1400 MPa-grade high-strength bolt steels with different Mo contents were used, and 82its composition is shown in Table. 1.” Where is Table 1???? The absence of this Table, which contains the different contents of Mo in the alloy, does not allow authors to deliver the aim of their study and thus destroy it.
- Page 3 from line 101 to line 105, the full definitions and full names for h, Eapplied, Ecorr, ico, ic, and ir must be mentioned in the text.
- Definitions of the elements for Eq. (10) should be written in the text. A reference number should be also inserted in the text referring to this equation.
- The potentiodynamic polarization curves of Figure 2 refers to E vs. log I curves for sample#1 and sample#2, which thickness of these samples? Why only one thickness is displayed in the Figure and not all samples? This is because the authors did not mention what was the area of each sample to be considered for corrosion test.
- Page 6, line 156, the authors again mentioned Table 1, which is not existed.
- Equation 11 needs a reference to be inserted in the text.
- The results part related to Figure 2, Figure 3, and Figure 4 are not enough and lakes interpretation.
- Where is Table 2???
- The Discussion part is too short and does not support the Results part.
Reviewer 4 Report
The manuscript is devoted to the study of the influence of Mo content on the hardness of high-strength steel. Unfortunately, the authors do not provide Tables 1 and 2 in the text of the article or in separate files (which could be downloaded). Because of this, we cannot fully consider this work, these data are very important. We hope that after providing these data, we will provide comments (in the next round), now we can say that the experiment is not clearly described in 2.1, there is no reference either.
Round 2
Reviewer 3 Report
The authors have improved their manuscript and performed most of my comments. The Abstract still needs to be improved and Eq. (10) must have a cited reference. If the authors do that the paper should be accepted for publication.
Reviewer 4 Report
Line 44-45 “These models demonstrate the following reaction, where hydrogen diffuses into metal in an acid solution [15]: ". The sentence is incorrect. Eq.1 represents hydrogen atom adsorption on the metal surface, but no diffusion into metal. The Eq.3. is corresponded to diffusion of hydrogen into metal.
Line 110-111 “Constant Eapplied to the samples at the hydrogen entry side.” The sentence is incorrect.
Fig 1, 6. The scheme assumes that the molybdenum atoms are nearby, and since there is no composition and method of preparing the alloy, this assumption contradicts the very idea of the first-principle calculation. Why is plane (110) used in Fig. 1, and (plane) 100 in Fig. 6? Is it of fundamental importance, or is it a typo?
Dilute solutions of acids and bases (0.05 and 0.2M, respectively) were used. It would be good to use the EIS-tests in order to make adjustments to the resistance of the solution itself.
The potentials should be given on the scale of a reversible hydrogen electrode in electrochemical studies.
